# Using Ground Penetrating Radar for Permafrost Monitoring from 2015–2017 at CALM Sites in the Pechora River Delta

**Maria Sudakova** [1,2,*], **Marat Sadurtdinov** [1], **Andrei Skvortsov** [1], **Andrei Tsarev** [1], **Galina Malkova** [1], **Nadezda Molokitina** [1] **and Vladimir Romanovsky** [1,3]

1   Earth Cryosphere Institute, Tyumen Scientific Centre SB RAS, 625000 Tyumen, Russia; mr_sadurtdinov@mail.ru (M.S.); agskvortsov@mail.ru (A.S.); tsarev.am1963@yandex.ru (A.T.); galina_malk@mail.ru (G.M.); molokitina.nadya@yandex.ru (N.M.); veromanovsky@alaska.edu (V.R.)
2   Geology Faculty, Lomonosov Moscow State University, 119991 Moscow, Russia
3   Geophysical Institute, University of Alaska Fairbanks, Fairbanks, AK 99775, USA
*   Correspondence: m.s.sudakova@yandex.ru

**Abstract:** This paper describes the results of ground penetrating radar (GPR) research combined with geocryological data collected from the Circumpolar Active Layer Monitoring (CALM) testing sites in Kashin and Kumzha in August 2015, 2016, and 2017. The study area was located on the Pechora River delta. Both sites were composed of sandy ground and the permafrost depth at the different sites ranged from 20 cm to 8–9 m. The combination of optimum offset and multifold GPR methods showed promising results in these investigations of sandy permafrost geological profiles. According to direct and indirect observations after the abnormally warm conditions in 2016, the thickness and water content of the active layer in 2017 almost returned to the values in 2015 in the Kashin area. In contrast, the lowering of the permafrost table continued at Kumzha, and lenses of thin frozen rocks that were observed in the thawed layer in August of 2015 and 2017 were absent in 2016. According to recent geocryological and geophysical observations, increasing permafrost degradation might be occurring in the Pechora River delta due to the instability of the thermal state of the permafrost.

**Keywords:** permafrost monitoring; water content measurements; CALM; GPR multifold; permafrost degradation





## 1. Introduction

The problem of global climate change has caused much concern worldwide. It is not trivial to estimate and predict the influence of climate change on the planet, and the combination of climatology, glaciology, oceanography, geomorphology, and geocryology knowledge is necessary to address this issue. Permafrost is a complicated multicomponent system that is ambiguously reactive to climate change [1]. The possibility of understanding the processes in permafrost areas and assessing the effects of a changing atmosphere on them may be accomplished by long-term monitoring. The upper part of the near surface, known as the active layer, is the most sensitive to climate change and human impacts [2]. The active layer thickness (ALT) and average temperature range in permafrost regions have been the subjects of geocryological monitoring and research in the context of geocryological forecasting [3,4].

Many scientists have observed the surface warming trend in Arctic regions over the last several decades (for example, [5,6]), which has confirmed the conclusion that there is a need for long-term permafrost monitoring.

The Circumpolar Active Layer Monitoring (CALM) project is a long-term international monitoring program that has provided active layer data from permafrost sites [7–9] since 1990. Within the program, ALT measurements are manually carried out at approximately 60 grids [https://ipa.arcticportal.org/products/gtn-p/calm] once a year by using a steel probe. The monitoring of CALM grids by integrated methods is presented in [4,10,11]. In

some cases, for example, at mountainous sites with rocky soils or sites with an ALT greater than 1.5 m, it is impossible to use mechanical probing (with a steel stick), but geophysical methods such as ground-penetrating radar (GPR) represent a viable alternative for ALT research [12,13].

In the framework of the CALM program, Brown proposed the use of GPR to determine the depth of thawing [7]. GPR can be successfully applied to the study of permafrost because ice, water, air, and soil have different electric permittivities [14]. The unfrozen/frozen soil boundary, which is the bottom of the active layer at the end of the thawing season, is a contrast interface for electromagnetic (EM) waves, and this fact is used to determine the thickness of the active layer [15]. The use of GPR allows not only the determination of the position of the permafrost table but also the identification of the permafrost structure, the presence of thawed lenses, and the moisture of the active layer [16–19].

To determine the depth of the bottom of the active layer and the velocity of EM waves, a multichannel (over 10 channels) observation georadar might be applied at one or two points per study area [15,19]. Often, the difference between the ALT measured mechanically and that calculated using GPR is typically no more than several centimeters; however, in individual cases, the error might be 50% or more [15]. In sites with shallow permafrost, mechanical probes can be used to determine the ALT. Then, the velocity of EM waves in the active layer is calculated by dividing the doubled layer thickness by the reflection arrival time. Usually, such measurements are point-like (rare, not made on a grid but at several points), and the average velocity value is used to determine the thickness of the active layer over the entire area. The errors can reach 20–25% [20]. However, there are examples of extremely small (2 cm) differences between the ALT estimated using single-channel GPR profiling and that measured manually [21].

The use of GPR for three-dimensional mapping of the bottom of the active layer was defined by Brosten [22]. Wollschlager showed that GPR can be used to map the depth of the bottom of the active layer and moisture content [23]. Gerhards et al. applied the multichannel GPR technique to measure the reflected wave travel time with three receivers located along the profile at different distances from the source [24]. Moreover, Westermann described a 5 week GPR campaign from August until September for monitoring the soil moisture content and the depth of the bottom of the active layer [12] using the same technique. Brosten published the results of the thaw depth under Arctic streams obtained during a summer campaign using GPR [25].

This paper considers the application of the GPR technique combined with geocryological data collected from the CALM sites in 2015, 2016, and 2017 on the island of Kashin (the CALM R24A site) and at the Kumzha site (the CALM R24A-2 site) to determine the active layer thickness and describe the three-year dynamics of the active layer. The main goal of our work is to show how changes in permafrost conditions over time are reflected in GPR data. Although only a three-year period is considered by our group, the changes in geophysical data are noticeable.

## 2. Study Area

Geocryological, meteorological, landscape, and geophysical (acoustic and georadar) surveys were carried out at the Kumzha site and at the Kashin Island site (Figure 1). Both monitoring sites were located in the northeastern European part of Russia within the Bolshezemelskaya tundra. The research area is formally located within the Nenets Autonomous District of the Arkhangelsk region. From the north, the sites are washed by Pechora Bay, which is a basin of the Barents Sea. The nearest large settlement is the city of Naryan-Mar, which is located 65 km southwest of the research area.

The research area is located in the southern tundra climatic zone. In the northern European region, areas of continuous, discontinuous, and sporadic permafrost were observed, and both open and through taliks are present here [26].

The study area was confined to the Pechora River delta. Around the study area, there are numerous branches of meandering rivers and swamp areas. Numerous distributary

channels and thermokarst lakes are widely spread throughout the vast floodplain of the Pechora River. The main warming effect is produced by the river and its distributary channels.

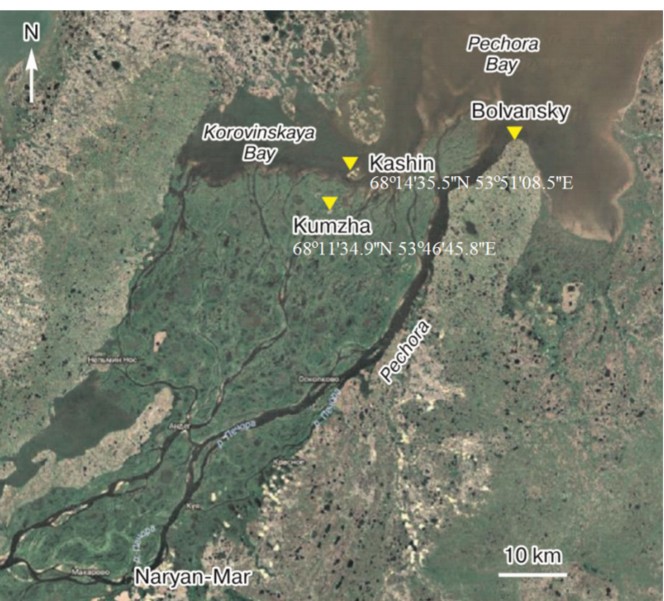

**Figure 1.** Locations of the Kashin (CALM 24A-1) and Kumzha (CALM 24A-2) study sites.

The Kashin and Kumzha sites are located in a sporadic permafrost region with permafrost temperatures of 0 to −1 °C. The results of temperature measurements in wells with a depth of 10 m revealed the unstable state of the permafrost in this landscape and under the present climatic conditions [6,27,28]. In the region, permafrost occurs sporadically in areas of natural undisturbed lichen tundra. In such areas, the thaw depth reaches 1.8–2 m in summer, and thawed soil freezes over the entire depth of the seasonally thawed layer, but the level of annual heat flux does not exceed 3 m.

The GPR surveys were conducted in August 2015, 2016, and 2017. Both sites were composed of sandy ground, which is a favorable condition for the propagation of a radio signal. In the Kashin area, the thickness of the active layer was no more than 150 cm. In the Kumzha area, the thickness of the active layer was greater than 150 cm. There was a continuous talik zone with a thickness of up to 8–9 m in the northeastern part of the study area. Interestingly, Kumzha was the exploratory drilling site of the Kumzhinskoye field at the end of the 1970s, which was responsible for the significant destruction of the natural landscape. In the areas around the CALM R24A-2 site, traces of old all-terrain tracks were observed.

### 2.1. Kashin Site (CALM R24A-1)

The Kashin geocryological station was founded in 2009. The station is located at the edge of the Pechora River delta on an island in Korovinskaya Bay. In the framework of the CALM program (R24A-1), thaw depth monitoring has been carried out in this territory since 2010. GPR studies at the Kashin station have been carried out continuously since 2015.

The Kashin CALM site was located on the I marine terrace with absolute elevations of up to 10 m. The surface of the terrace is slightly undulating and is separated by lake basins and runoff channels.

According to geophysical electrotomography data, the thickness of the permafrost on the island does not exceed 30–40 m [29]. Although permafrost is sporadic in the region, the island is composed of frozen sands. At the top part of the geologic profile in some sections, peat of various thicknesses is found (from 0 to 30 cm at the site).

Previous studies have shown [30] that the depth of thaw at the Kashin site (CALM R24A-1) is related to the lithology, particularly the thickness of peat. Within the Kashin site

in sandy patches with a peat thickness less than 10–12 cm where tundra vegetation spreads, the ALT ranges from 70 to 140 cm, but at swampy sites and sites with a peat thickness from 15 to 30 cm, the ALT does not exceed 70 cm.

### 2.2. Kumzha Site (CALM R24A-2)

Studies at the Kumzha site have been conducted since 2013, and the CALM 24A-2 site was founded in 2014. GPR studies in the Kumzha area have been carried out since 2015.

The site is located on the remnant of the I alluvial-marine terrace in the Pechora River delta. The geological profile consists of dense sands with horizons of ironization and oil contamination. At the end of the 1970s, an exploratory drilling site was located on the remnant, and as a result, the natural landscape was significantly disturbed. At present, the Kumzha area still shows traces of all-terrain tracks, clearing, excavations, bourocks, etc. However, after exploratory drilling was carried out and the integrity of the vegetation cover and the surface topography were disturbed, the thawing depth began to increase gradually. At present, lowering of the top layer of permafrost is occurring on most of the outlier of the I alluvial-marine terrace [27,29,31]. There is no reliable information on the permafrost depth.

The CALM R24A-2 site is located in a relatively undisturbed area with preserved natural vegetation, such as lichen tundra, replacing the sparse birch forest that is located 100 m from the edge of the quarry. However, traces of old all-terrain tracks are present.

### 3. Methods

The research methods used at the Kashin and Kumzha CALM sites from 2015 to 2017 are presented in Table 1. Over the years, GPR common-offset profiling and multifold (MF) common receiver gather (CRG) measurements, seismic measurements, ALT measurements with a probe, water content measurements by weight, and temperature measurements at the surface and in the boreholes were carried out. Observational temperature boreholes were drilled to a depth of 10 m in Kashin in 2012 (3 wells) and in Kumzha in 2014 and 2016 (4 wells). Figure 2 shows a sketch of the geophysical surveys and measurements of the thaw depth with a probe at CALM sites. As shown in Figure 2, we used letters and numbers to name CALM grid points and geophysical profiles. This article focuses mainly on GPR investigations; therefore, a detailed description of all other methods is beyond the scope of this article.

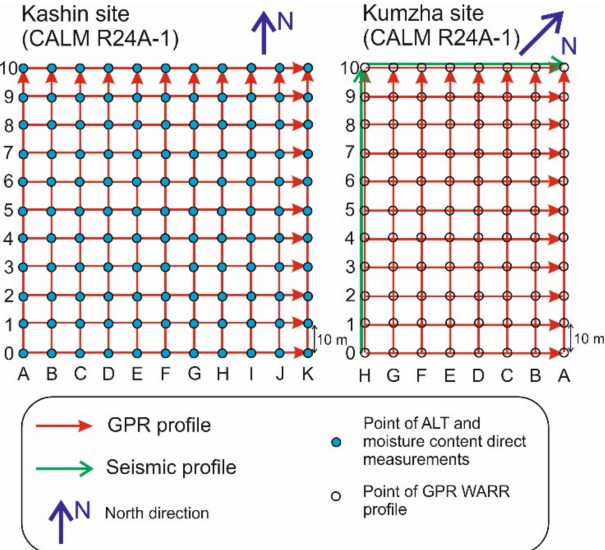

**Figure 2.** Sketch of the GPR and seismic transects and points of direct measurements.

**Table 1.** Research methods at the Kashin (CALM 24) and Kumzha (CALM 24A) sites.

| Year | Kashin | Kumzha |
|------|--------|--------|
| 2015 | GPR common-offset profiling<br>Manual probing of the ALT<br>Temperature measurements<br>Thermostatic weight method for<br>moisture measurements | GPR common-offset profiling<br>Seismic refraction method and common depth point method<br>Temperature measurements |
| 2016 | GPR common-offset profiling<br>Manual probing of the ALT<br>Temperature measurements | GPR common-offset profiling<br>Seismic refraction method and common depth point method<br>Temperature measurements |
| 2017 | GPR common-offset profiling<br>Manual probing of the ALT<br>Temperature measurements | GPR common-offset profiling<br>Seismic refraction method and common depth point method<br>Temperature measurements<br>GPR multifold measurements |

To carry out GPR research, a Zond-12e georadar (Radar Inc., Riga, Latvia) was used. GPR profiling was carried out with 300 MHz shielded bow-tie antennas. The GPR acquisitions were carried out along two orthogonal directions. The profiles crossed CALM grid points, and the distance between parallel profiles was 10 m. The distance between the points on the profile line was no more than 10 cm. The alignment of the points on the profile lines was carried out using marks that were placed every 10 m at the intersection profile lines. The recording parameters were chosen based on the depth of the permafrost table.

The data processing in all sections consisted of the following procedures: scaling GPR data by marks, time zero correction, bandpass filtering (Ormsby filter), spherical spreading recovery, 2D filtering (antenna ringdown subtraction), and predictive deconvolution of the data from the Kashin site. After applying these procedures, continuous reflection events were picked with verification at profile crossing points, and the construction of the depth profiles and maps was also carried out.

In addition to GPR profiling, to determine the velocity of EM waves and accurately determine the depth of boundaries, MF GPR measurements were collected in 2017 at the Kumzha site. GPR CRG measurements were carried out at each point of the CALM grid using two separate 150 MHz dipole antennas. We used the CRG data acquisition geometry and oriented the copolarized antennas with their broadsides toward each other (as described in [32]). The maximum offset was equal to 20 m, while the offset increment was 1 m. In 2015 and 2016, the apparent velocities (the velocity in the layer from the surface to the boundary assuming the rays are direct without refraction) within the section were determined using diffraction hyperbolas. Seismic measurements were carried out on two profiles at the Kumzha site using refracted waves with P- and SH-waves (see Figure 2).

In the Kashin area, measurements of the thaw depth were carried out using a metal probe at each grid point of the CALM site, with a grid size of 100 × 100 m and a grid interval of 10 m. The metal probe had a cross-section of 8–10 mm and a length of 1.5 m and was a pointed metal wire with a T-shaped crossbar at the top. The water content of seasonally thawed soil was measured at each grid point of the CALM site on Kashin Island. A comparison of the water content maps measured by the thermostatic weight method at a depth of 20 cm and the volumetric moisture maps of the entire active layer calculated according to GPR data and based on field season results gathered at the Kashin site in 2015 was published in a previous article [13].

Moreover, throughout the year, monitoring of the air temperature, surface, active layer, and permafrost at the sites and in wells located near the CALM sites was conducted in the Kashin and Kumzha areas using automatic loggers and temperature sensors [31,33].

## 4. Results

### 4.1. The Kashin Site (CALM R24A-1)

The typical GPR data obtained from one of the profiles (marked with symbols I0-I10 in Figure 2) at the Kashin site in 2015–2017 are presented in Figure 3. The data were processed according to the methods described above. A high-amplitude reflection in the GPR data clearly corresponds to the bottom of the active layer or the top of the permafrost (shown by a black line). Next to the reflection from the active layer bottom, there are multiple reflections, as shown by the thin dotted line. The dashed line shows the time corresponding to a depth of 150 cm with a constant velocity on the profile of 7 cm/ns. The permafrost top reflection in the I0–I10 profile was above the dashed line in 2015, lower than the dashed line throughout most of the GPR section in 2016, and close to the 2015 reflection position in 2017. Such changes in the value of the reflected wave time were associated with either an increase in the depth of the active layer, an increase in the effective velocity of the EM wave in the seasonal thaw layer (decrease in moisture), or a combination of both factors acting simultaneously.

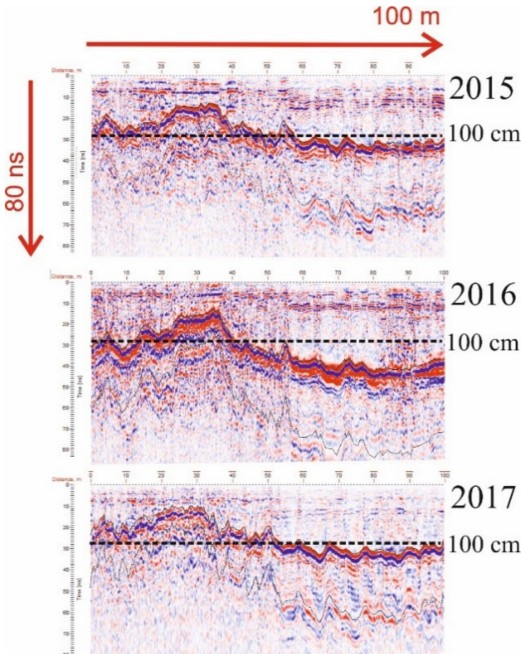

**Figure 3.** GPR data obtained in the Kashin area (CALM R24A-1) in 2015, 2016, and 2017 along the I0–I10 transect. The black line is the reflection event of the active layer bottom; the dotted black line is the multiple reflection events of the active layer bottom. The dashed line corresponds to a depth of 100 cm and a constant speed of 7 cm/ns.

The ALT at the CALM R24A-1 site during the research period did not exceed 150 cm; therefore, at the points of the grid, it was measured with a probe. Figure 4 shows a map of the seasonal thaw depth measured with the probe in 2015–2017. The study area was characterized by an irregular thaw depth in all years. According to direct measurements, the ALT varied from 37 cm to 117 cm in 2015, from 46 cm to 150 cm in 2016, and from 34 cm to 119 in 2017. It was established that in 2016, the minimum and maximum values of the ALT were 25% greater than those in 2015 and 2017, but in 2017, the thaw depth was the same as that in 2015.

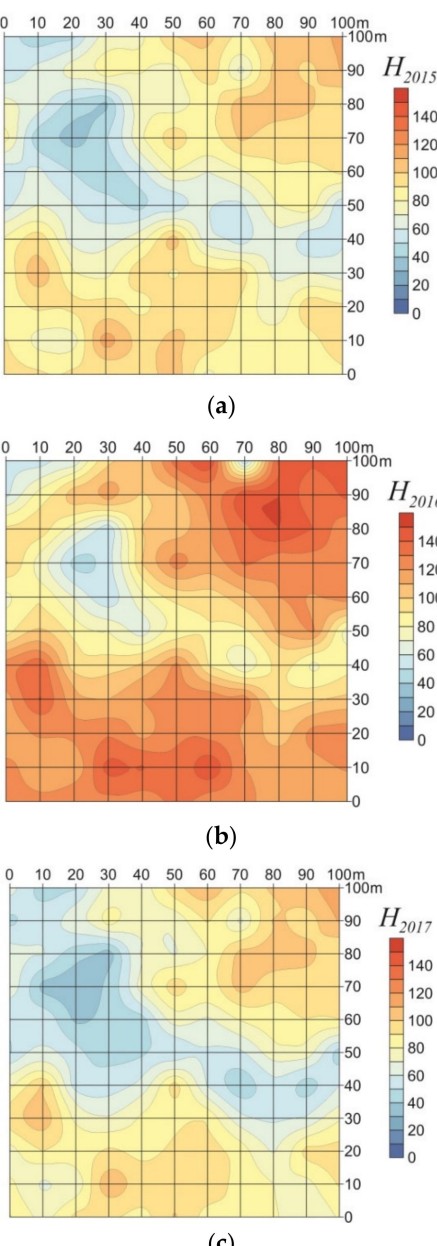

**Figure 4.** The thickness of the active layer H (cm) in the Kashin area (CALM R24A-1) in 2015 (**a**), 2016 (**b**), and 2017 (**c**).

According to the known values of permafrost thickness at the points of the CALM grid and the reflection arrival time, the velocity of EM waves was calculated. The average velocity values were approximately 5.7, 7.2, and 6 cm/ns in 2015, 2016, and 2017, respectively. Moreover, significant data scattering (more than two times) was evidenced in all years. The velocity values were used to calculate the dielectric constant, which was used to determine the volumetric water content in the active layer according to GPR profiling in 2015, 2016, and 2017. Unfrozen sediments at the Kashin site consist of nonsaline sand and peat, so we did not expect significant values of electrical conductivity and used a formula for dielectric properties without losses to calculate the dielectric constant from velocity. Nevertheless, to verify this approximation, we estimated radar energy attenuation based on the amplitudes of the reflected signal from the permafrost table at different depths. The attenuation ranges from 0.06 m$^{-1}$ to 0.1 m$^{-1}$, and conductive losses should not have much impact on EM velocity in this case, so the use of a lossless approximation is appropriate.

For the water content calculation, we used Topp's formula [34]. Volumetric moisture maps of the active layer collected with a step of 10 m are shown in Figure 5. The trend of changes in moisture of the active layer in 2015–2017 at the Kashin site was the same as that of the thickness of the active layer: according to GPR data in 2015, the volumetric moisture increased from 30 to 95%; in 2016, the moisture values were significantly lower at 10–60%; and in 2017, the moisture increased but did not reach the values that were achieved in 2015, with a minimum volumetric moisture value of 20% and a maximum value of 70% in 2017.

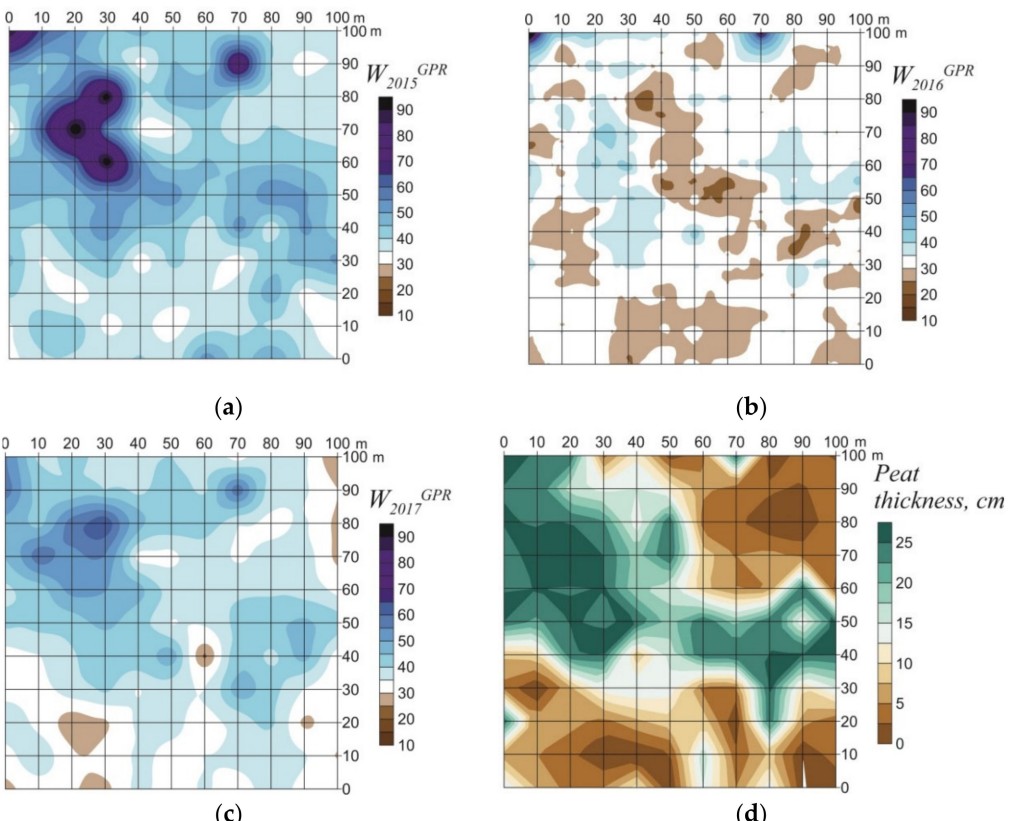

**Figure 5.** The volumetric water content of the active layer W $^{GPR}$ (%) calculated according to the GPR data at the Kashin site (CALM R24A-1) in 2015 (**a**), 2016 (**b**), and 2017 (**c**). (**d**) shows the thickness of the peat.

It should be noted that the moisture maps in 2015 and 2017 are analogous to each other (Figure 5a,b): a characteristic region of high moisture content crosses the site from the northwest to the south-southeast on both maps. This phenomenon can be explained by the fact that this area is covered by peat, which contains much more free water than do mineral soils (Figure 5c).

### 4.2. The Kumzha Site (CALM R24A-2)

The permafrost lies at a depth of more than 1.8 m over almost the whole site. Due to this fact, it is impossible to determine the position of the permafrost table using a probe. To solve this problem, a series of geophysical methods (GPR and seismic measurements) was applied in the study area. Typical GPR data obtained from the H0-H10 transect at the Kumzha site from 2015–2017 are shown in Figure 6a (common-offset profiling) and Figure 6b (MF CRG method). The data are shown after the processing described above in the Section 3. A reflection event off the top of the permafrost might be noticeable in the data. The amplitude and wavelength of the reflection change little throughout the profile. On the left part of the profile (south side of the site), a reflection occured at 80–100 ns, and by the end of the profile (the northern part of the site), it had decreased to approximately

250 ns. Differences in the depth of the top of the permafrost suggest that a through talik occurs in the northern half of the site [33].

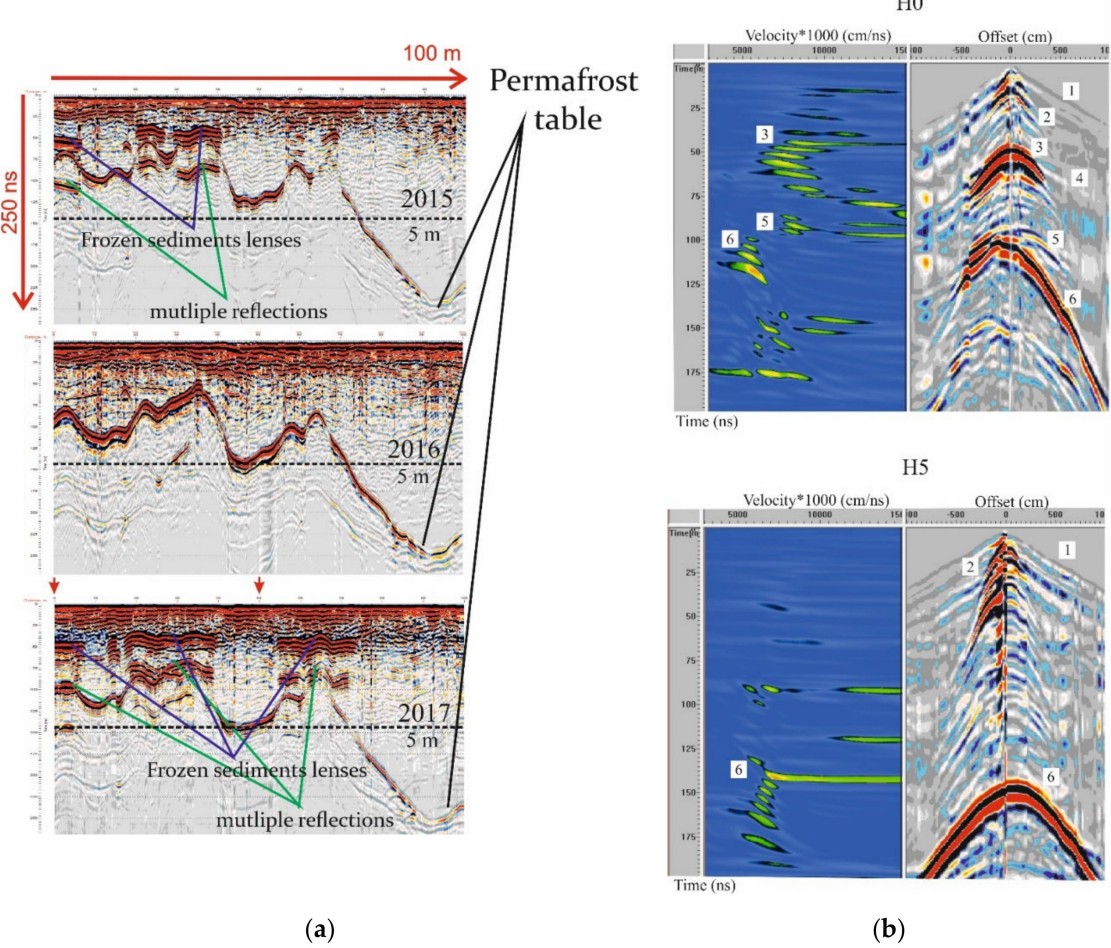

**Figure 6.** GPR data collected at the Kumzha site (CALM R24A-2) in 2015, 2016, and 2017 from the H0-H10 profile (**a**) and MF GPR data collected from the H0 and H5 points in 2017 (**b**). The dashed line corresponds to a depth of 5 m with a constant velocity of 7 cm/ns. The red arrow shows the point where the MF GPR data were collected (on the right) in 2017. For the MF GPR data (**b**), the numbers indicate the following waves: (1)—an airwave, with a velocity of 30 cm/ns, (2)—a direct wave in dry sand, with a velocity of 10–12 cm/ns, (3)—a reflected wave from the top of a frozen layer (lens) within unfrozen sand, with a velocity of 8 cm/ns, (4)—refracted reflection, with a velocity of 30 cm/ns, (5)—multiple reflections from the top of the unfrozen inner layer, and (6)—the reflected wave from the top of the permafrost, with a velocity of 5–6 cm/ns.

In the GPR data obtained in 2015 and 2017, high-amplitude reflections are located above the permafrost table. Below, at double and even triple times, multiple reflections are visible. This boundary is associated with a frozen layer, which does not thaw over the summer period. In the data in Figure 6, the time between the reflection from the top and the bottom of the layer reaches 10 ns, which corresponds to a thickness of approximately 75 cm considering the EM velocity in frozen sand (approximately 15 cm/ns). To detect the real thickness of this layer, drilling was performed by our group near the CALM grid at a point with similar GPR data: the thickness there was approximately 10 cm. This mismatch could be explained by the warming effect of intervention (drilling, warm surface airflow) and the unstable state of this layer.

On the GPR profile obtained along the H0-H10 profile in 2016, there were no reflections from the thin layer of frozen sand in the thickness of thawed rocks since this layer was completely thawed due to an abnormally warm summer.

At a distance of 20 m from the beginning of the profile at approximately 50 ns, a diffraction hyperbola was visible in all GPR patterns. The velocity along this hyperbola was approximately 7 cm/ns. The scatter of velocity values at the Kashin site is similar to that at the Kumzha site, indicating that one average velocity value observed from diffraction hyperbolas is not sufficient for the determination of the position of the permafrost table. Therefore, the MF GPR CRG method was added to collect data from all points of the CALM 24A-2 site in 2017.

Figure 6b shows the data collected via the CRG method. These data were recorded at points H1 and H5 along the profile H1-H10 (see Figure 2), and the points are shown on the GPR profile obtained in 2017 by red arrows (Figure 6a, bottom left). The gathers were recorded while the receiver antenna was located at a certain point and the source antenna was moving along the profile. Figure 6b shows GPR MF data. The main difference between the data obtained at points H0 and H5 in the presence or absence of a reflection from the inner frozen layer: there is a reflection at point H0 but not at point H5. This difference can be seen in the common-offset data (Figure 6a). According to this, waves of different classes and with different velocities are distinguished, and maximums on the processed display correspond to some of the reflected waves. The numbers indicate the following: (1)—an airwave, with a velocity of 30 cm/ns, (2)—a direct wave in dry sand, with a velocity of 10–12 cm/ns, (3)—a reflected wave from the top of a frozen layer (lens) within unfrozen sand, with a velocity of 8 cm/ns, (4)—refracted reflection, with a velocity of 30 cm/ns, (5)—multiple reflections from the top of the unfrozen inner layer, and (6)—the reflected wave from the top of the permafrost, with a velocity of 5–6 cm/ns. The velocities of waves detailed above are apparent velocities. The velocity of the reflected wave from the top of the frozen layer was less than the direct wave velocity, and its hodograph was not asymptotic to the direct wave hodograph because between the air surface and frozen layer, there is a water table. The reflected and refracted waves from the water table were visible in the data, and the higher point of the reflected wave hyperbola (t0) was located at approximately 10 ns.

The shape of the permafrost table at Kumzha site is not horizontal and the processed version of all the CRG data from Kumzha site is not informative; therefore, to determine the velocity of EM waves down to the depth of the top of the permafrost, we used hyperbola fitting, setting the velocity in the layer from the surface to the boundary, t0 and the angle of inclination of the reflective layer.

The maps of the permafrost top at the Kumzha site prepared according to GPR data are shown in Figure 7a,b. The depth map $H^1_{2017}$ was constructed based on the results of the MF and common-offset GPR methods using velocity changes in the lateral direction. The depth map $H^2_{2017}$ was created based on only common-offset GPR profiling. A constant velocity of 7 cm/ns was used to recalculate the arrival time at depth for $H^2_{2017}$. Figure 7c,d presents maps of the absolute and relative differences in depths. The error in determining the depth of permafrost could reach 150 cm if the lateral velocity changes are not taken into account.

Maps of the permafrost depth based on the GPR data collected from the CALM R24A-2 site in 2015–2017 are shown in Figure 8. According to the GPR data in the southern part of the Kumzha area, the permafrost is located at a depth of 1.50 to 2.50 m. In winter, the thaw layer completely freezes, as evidenced by the data from the thermal observation wells [33]. In the northern part of the research area, technogenic disturbances existed. For example, there were areas deprived of vegetation as a result of the repeated passage of vehicles. The top of the permafrost gradually receded from a depth of 3 m to a depth of 9.5 m. This finding indicated that there was an open talik because intensive cooling and seasonal freezing of the ground in winter might occur to a depth of only 2.5–3 m in the absence of vegetation. In 2016, there was an increase in the time of the reflection from the permafrost table by 11–17 ns. Considering a constant velocity of 7 cm/ns, the permafrost table in the research area (the ALT in the southern part of the site) was 40–60 cm deeper than that in 2015. In 2017, the permafrost table was approximately 50 cm higher on average

than that in 2016. In addition, local decreases and increases in the permafrost table depth were observed. Concerning this result (permafrost table lowering), the lack of EM wave velocity data in 2015 and 2016 should be taken into account.

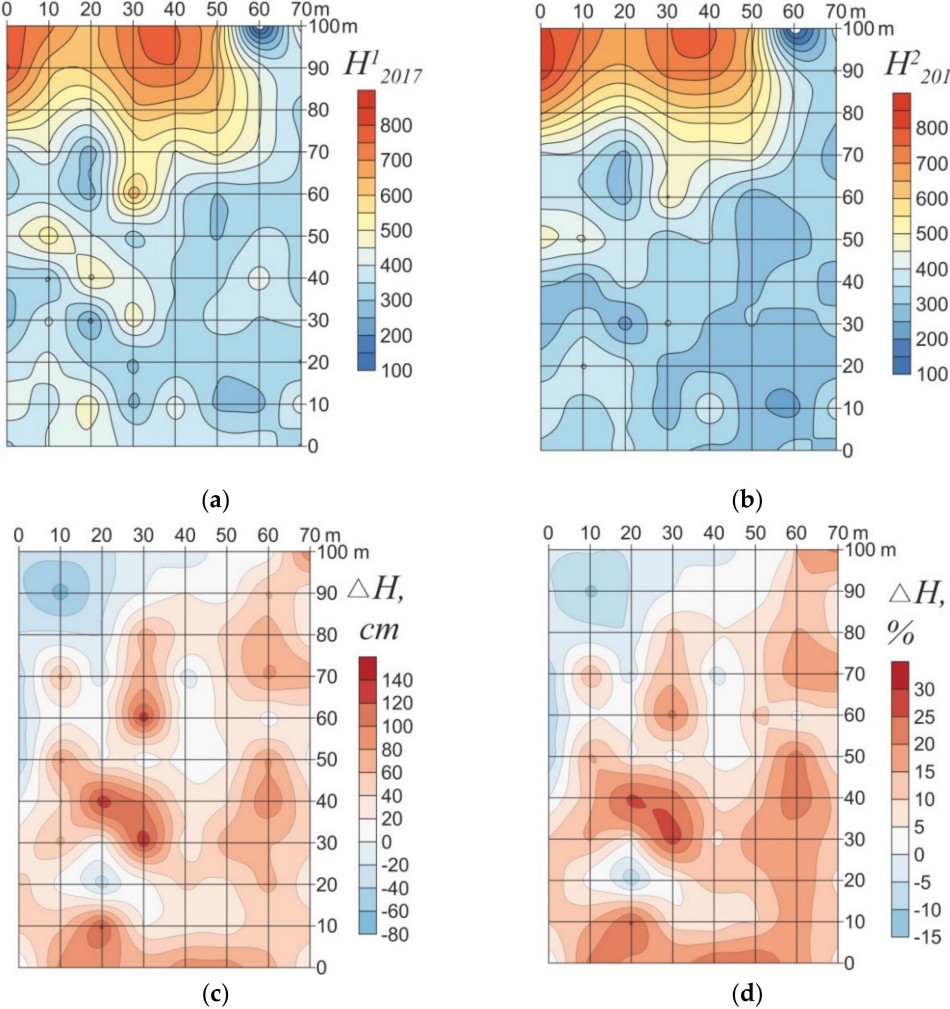

**Figure 7.** Depth ($H$) of the permafrost table at the Kumzha site (CALM R24A-2) in 2017: (**a**) $H^1_{2017}$—according to the GPR data taking into account the changes in velocity at the site, (**b**) $H^2_{2017}$—without taking into account the velocity measurements at the site (velocity 7 cm/ns determined according to the hyperbole on one of the profiles), (**c**) $\Delta H$—absolute difference between the depths $H^1_{2017}$ and $H^2_{2017}$, cm, (**d**) $\Delta H$—relative difference between the depths $H^1_{2017}$ and $H^2_{2017}$,%.

Figure 9 presents maps of the differences in the permafrost depths in 2015–2016 and 2016–2017. The increase in the maximum thaw depth (1 m or more) was distributed in the research area according to the characteristic lineaments in 2016. A satellite image of the research site is shown in Figure 9b. As shown in the picture, there are two types of structures. The first type is natural polygonal structures inherited from ice wedges, and the second type is ruts from transport vehicles; these structures are shown as blue dots and red lines, respectively. The map of the differences clearly shows (Figure 9b) that the territories where an abnormal increase in depth might be observed were in good agreement with the position of the polygonal structures. In 2017, the top of the permafrost continued to recede, but there was no deviation between the lineaments (track and ice-wedge patterns) on the difference map.

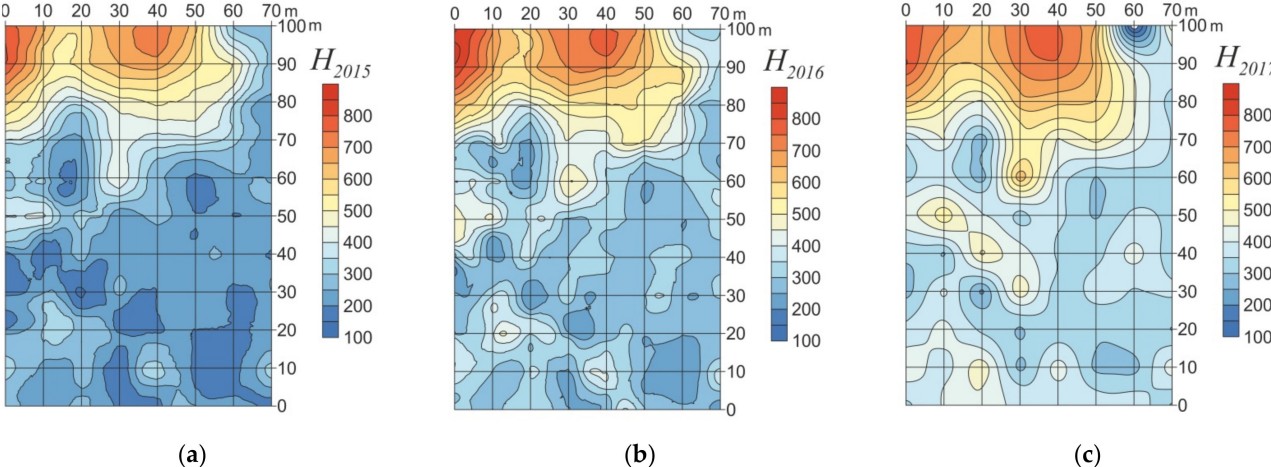

**Figure 8.** Depth of the permafrost table (H) at the Kumzha (CALM R24A-2) site in 2015 (**a**), 2016 (**b**), and 2017 (**c**) according to GPR data.

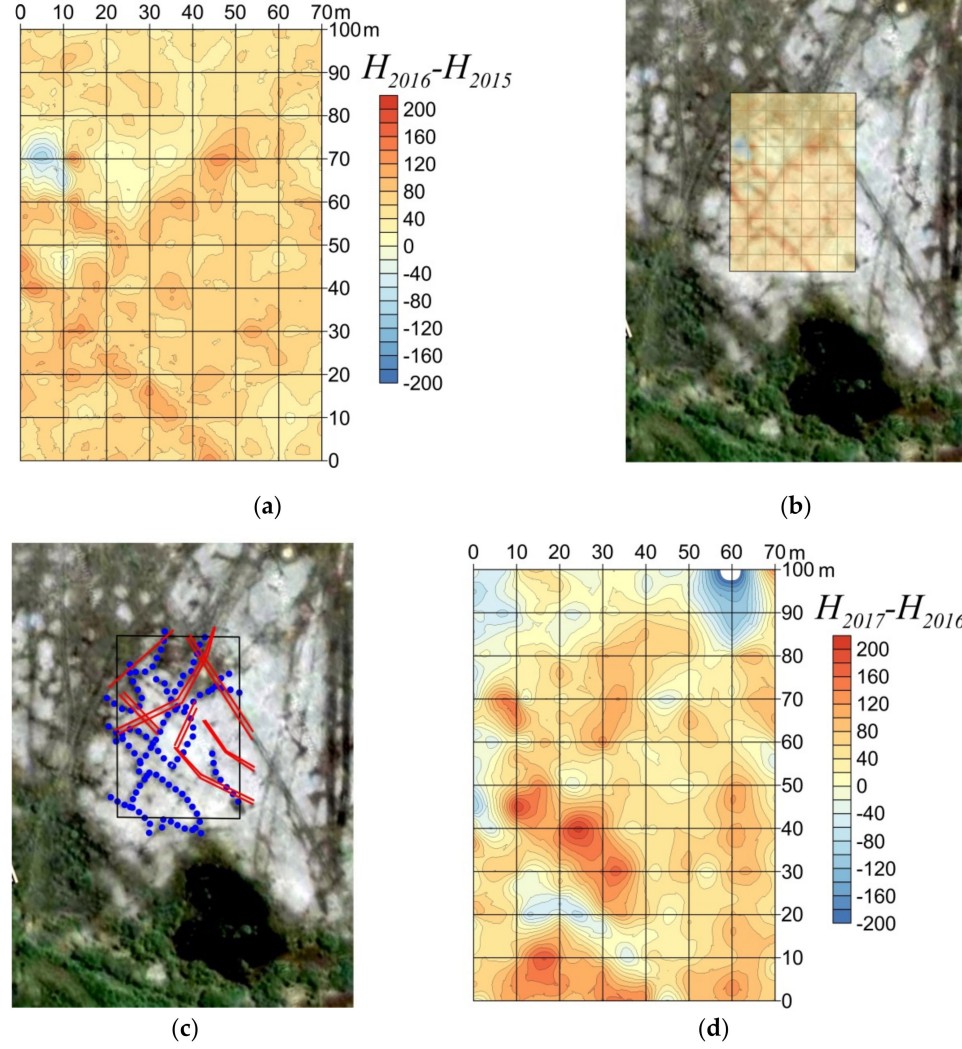

**Figure 9.** (**a**) The differences in the depth of the permafrost table at the Kumzha site (CALM R24A-2) in 2016–2015 according to GPR data, (**b**) the difference in the 2016–2015 map is overlaid on a satellite image, (**c**) satellite image with former roads (red) and natural polygonal structure borders (blue) highlighted, (**d**) difference in the depth of the permafrost table at the Kumzha site from 2017–2016 according to GPR data.

Thus, the position of the maximum difference was probably due to the different nature of the change in the water content within and nearby polygonal structures and to the composition of the soils along the borders of these polygon structures. Both the water content and soil composition directly affect the velocity of EM waves in the thawed layer.

The anthropogenic impact had practically no effect on the geocryological conditions of the site. Thus, anthropogenic disturbances such as roads generally do not appear on the maps of the differences in the permafrost table depth over the years.

## 5. Discussion

At the Kashin site, permafrost is shallow, and the ALT is no more than 150 cm. In such cases, for CALM purposes, ALT could be measured manually at the grid points, and geophysical methods could be used to obtain additional information about other features or properties of the active layer, such as the water content. Our results showed that in the 100-by-100 m square of the Kashin CALM site, the water content changes significantly, and these changes are reflected in the EM velocity field: within this site, the EM velocity differs more than two times laterally. This variability can lead to errors in ALT estimates if only common-offset GPR is used.

The results show that both the geometry of the permafrost table and the properties of the active layer can vary significantly vertically, laterally, and over time. The GPR method was used to monitor not only geocryological boundaries but also the water content in the active (thawed) layer. Water content (as well as the velocity of EM waves) was shown to vary significantly within a square site with sides of several tens of meters. Using a constant velocity for EM waves in processing GPR data, the depths of geological boundaries might be determined incorrectly within an inhomogeneous site.

The prevalence of errors generated a need to set the transmitter and receiver separately from each other (using the MF method) to determine the depth of the top of the permafrost layer. Without considering the horizontal velocity gradient, the error in estimating the depth of the top of the permafrost layer was approximately 1.5 m in the Kumzha area. In our case, MF GPR at every point of the CALM grid at Kumzha in 2017 was performed by our group. Unfortunately, in 2015 and 2016, only a common-offset GPR survey was performed, so the depth of the top of the permafrost layer could only be approximated in these years. A comparison of the permafrost depth maps plotted with and without taking into account the EM velocity gradient in 2017 (Figure 7) confirmed this. At the same time, there was an increase in the arrival time of the wave reflected from the top of the permafrost layer at the Kumzha site in 2016 compared to 2015 as well as in 2017 compared to 2016, which also indicates the increasing depth of this boundary. The fact that the maximum arrival time of the permafrost table increase coincided with former ice wedges indicates that the maximum changes affecting the arrival time of the wave at the site are occurring along these boundaries. Additionally, the maps of the difference in the position of the permafrost top in 2016 and 2017 show that changes in both depth and velocity (water content) are possible.

Table 2 shows the main results of the GPR investigations at the CALM 24A-1 and CALM 24A-2 sites. The average surface temperature was calculated based on the results of the measurements collected near the sites. A correlation between the changes in the average annual temperature and the thaw depths and moisture was observed at the Kashin site. The average thickness of the seasonally thawed layer obtained by probing was 30% higher in 2016 than in 2015 and 2017. In 2017, the average thaw depth almost reached the value of 2015. The average volumetric water content obtained by GPR in 2016 was 30% less than that in 2015 and 2017. The abnormally hot conditions in 2016 caused drying of the active layer at the site.

**Table 2.** Summarized monitoring results at the CALM 24A-1 and CALM 24A-2 sites.

| Time Scale | Average Temperature at the Surface, °C | Field Season | Kashin (CALM R24A-1) | | Kumzha (CALM R24A-2) Depth of the Permafrost Table, cm |
| --- | --- | --- | --- | --- | --- |
| | | | ALT [1], cm | Volumetric Water Content of AL [2], % | |
| 14 August 2014–13 August 2015 | −1.6 | August 2015 | 80 | 44 | 350 |
| 14 August 2015–13 August 2016 | −0.2 | August 2016 | 109 | 32 | 380 |
| 14 August 2016–13 August 2017 | −1.6 | August 2017 | 76 | 39 | 400 |

[1] ALT, active layer thickness; [2] AL, active layer.

For the Kumzha site, the general result differed from the result for the Kashin area. Thus, the measurable indicators reached the initial value at the Kashin site, but the subsidence of the permafrost layer continued but slowed in the Kumzha area in 2017. The difference between the average depths of the permafrost table in 2016 and 2015 was 30 cm, and the average depth of the permafrost table in 2017 was 20 cm greater than that in 2016 at the site.

## 6. Conclusions

Depending on the thickness of the active layer, GPR was used as the main or additional method for studying and monitoring the near-surface conditions. In this work, the top of the permafrost ranges from depths of 30 cm to 8–9 m, and the reflections were visible according to the GPR data collected with a 300 MHz antenna.

At both sites, there was a correlation between the changes in the average annual temperature and the thaw depth and water content in the Kashin area. According to direct and indirect observations, after the warm conditions in 2016, the ALT and the water content in 2017 almost returned to the values in 2015 in the Kashin area. In contrast, the lowering of the permafrost table continued at Kumzha, and lenses of thin frozen ground that were observed in the thawed layer in August of 2015 and 2017 were absent in 2016.

The unstable thermal state of discontinuous permafrost in the southern tundra zone is caused by climate warming. According to the geocryological and geophysical observations of recent years, increasing permafrost degradation might be seen in the Pechora River delta due to the instability of the thermal state of permafrost. The same results were also described in [28,29,31,33,34].

**Author Contributions:** Conceptualization, M.S. (Maria Sudakova) and V.R.; fieldwork, M.S. (Marat Sadurtdinov), A.S., A.T. and G.M.; writing, M.S. (Maria Sudakova) and N.M.; review and editing, M.S. (Marat Sadurtdinov), A.S., G.M. and V.R.; data processing, A.T. and M.S (Maria Sudakova); visualization, M.S. (Maria Sudakova) and A.S.; supervision, V.R.; project administration, M.S. (Marat Sadurtdinov); funding acquisition, G.M. All authors have read and agreed to the published version of the manuscript.

**Funding:** This work was supported by the international projects CALM and TSP, RSF grant 16-07-00102.

**Conflicts of Interest:** The authors declare no conflict of interest.

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
