# Peer review of "Using Ground Penetrating Radar for Permafrost Monitoring from 2015–2017 at CALM Sites in the Pechora River Delta"

_remotesensing, doi:10.3390/rs13163271_

Round 1
Reviewer 1 Report
The paper is devoted to the topic of permafrost monitoring, which is relevant in connection with climate changes and the degradation of permafrost noted in many places, which can lead to catastrophic consequences. The novelty of the work is that the GPR method is used for monitoring, being highly sensitive to the position of the boundaries between water-saturated and frozen rocks.
The GPR results presented in the paper demonstrate the sensitivity of the method to the position of the bottom of the seasonally thawed layer, the presence of lenses of frozen rocks, and to the position of the permafrost top. According to the results of repeated measurements carried out over three years, one can see the change in the position of all these boundaries. In particular, an increase in depth to the permafrost top was recorded in one area.
In addition, the paper presents the application of sophisticated GPR techniques. To increase the reliability of interpretation, in addition to the standard scheme, sounding at one site was performed using different offsets between the source and the receiver of electromagnetic waves. In addition, for the interpretation the authors used data from the direct determination of the depths of the boundaries using a probe, as well as temperature and seismic studies. This made it possible to reliably determine the position of the boundaries of the layers and their properties, in particular humidity, as well as estimate their changes in time.
Thus, the paper presents important new scientific results. The research objectives, methodology and results obtained are well substantiated. The article is well structured and contains all the necessary sections. I don’t have any negative comments on the article. In my opinion, it can be published as presented.
Author Response
Dear Reviewer,
We truly appreciate your high ratings of our article.
We have carefully revised the manuscript, and we truly think this article is suitable for Remote Sensing.
Many thanks again!
Reviewer 2 Report
This manuscript used geophysical method to monitor the permafrost change at CALM sites in the Pechora River delta. The results can help us to understand the spatial variation of permafrost within a short distance, and its change with the extreme climatic event, as well as the availability of ground penetrating radar for permafrost detection. Some specific comments of the manuscript are as below:
General comments
This study is of important significance for permafrost research. However, the Introduction section in the manuscript does not differentiate it from the previous studies, only a list of literatures there. In the Results section, please mainly describe the findings of this study, some descriptions seem like “Methods” of “Discussion” should be moved to the corresponding section.
This study mentioned two GPR velocity detection methods, CRG and MF. Do they have any difference? If they do, please describe in detail with sketch map. If not, please use a single name to avoid confusion for readers.
There are two methods for field survey detection of ALT, GPR and mechanically methods, what are the differences between them at the two study sites? What is the difference between the measured soil moisture and the GPR calculated moisture at the Kashin site?
The language and logic of the manuscript still needs to be greatly improved. For example, some sentences are too long to read, please split into short sentences, and some incorrect spelling should be revised carefully.
Minor concerns
L49: GRP --> GPR, the same wrong spelling also can be found in other lines, please check them carefully
L131: Thou –> Though?
L159: thickness --> ALT? if so, an abbreviation of ALT will be better to keep consistency with other description (e.g. thickness, thickness of the active layer, or thickness of the seasonal thaw layer) in the manuscript
L172: permafrost top layer --> permafrost table?
L176: Kasin or Kashin?
L194-196: here mentioned the seismic measurements, but its result does not find in the following part.
L280: B represents Figure6 b?
L336-L339: In the first section (Line336-340), the authors said the velocity was determined through hyperbolas fitting for the not horizontal permafrost table layer, but in the following section (Line341-343) only "multifold" velocity detection method was mentioned, this is confused for readers, and how the authors acquired the changing velocity along a profile
L343: miltifold --> multifold
L414, L419, L428: What is the difference among permafrost top layer, permafrost table and permafrost roof?
L453-454: the correlation only found in Kashin site or both site?
Author Response
Dear Reviewer,
Thank you for your attention to our article.
We tried to properly answer all of your questions and correct all of our mistakes marked by you.
A point-by-point response is attached.
We hope that you will find all of our corrections suitable.
This study mentioned two GPR velocity detection methods, CRG and MF. Do they have any difference? If they do, please describe in detail with sketch map. If not, please use a single name to avoid confusion for readers.
Multifold (MF) is the name for the different source-receiver position geometries. There are some common MF geometries, such as the common middle point (CMP), common source gather (CRG) and common receiver gather (CRG). All of them are described in the article of Forte: Forte E. and Pipan M., 2017, Review of multi offset GPR applications: data acquisition, processing and analysis, Signal Processing, 132C, 210-220, 10.1016/j.sigpro.2016.04.011.
Therefore, the MF is the general term, and CRG is the specific term.
There are two methods for field survey detection of ALT, GPR and mechanically methods, what are the differences between them at the two study sites?
We used mechanical manual probing only at the Kashin site because at the Kumzha site, the permafrost table is too deep for manual probing.
Regarding the details of the techniques, there were no differences in GPR techniques employed at the two study sites.
If the question is about differences in the permafrost table depth obtained by GPR and manually probing, there are no differences either. We used ALT obtained at points on the CALM grid (10x10 m) to calculate the EM velocity and interpolate it for the whole CALM site.
Our aim was to use both direct and indirect methods together.
What is the difference between the measured soil moisture and the GPR calculated moisture at the Kashin site?
These two moisture values could be compared only at the quality level because directly measured soil moisture corresponds to the 20 cm depth, and moisture obtained by GPR is an integrative characteristic of the whole active layer (depths from 0 m to the permafrost table).
We calculated a correlation coefficient of approximately 0.6.
The language and logic of the manuscript still needs to be greatly improved. For example, some sentences are too long to read, please split into short sentences, and some incorrect spelling should be revised carefully.
The AJE editing certificate is attached below.
Minor concerns
L49: GRP --> GPR, the same wrong spelling also can be found in other lines, please check them carefully
Done.
L131: Thou –> Though?
Corrected.
L159: thickness --> ALT? if so, an abbreviation of ALT will be better to keep consistency with other description (e.g. thickness, thickness of the active layer, or thickness of the seasonal thaw layer) in the manuscript
The term “thickness” has been changed to “active layer thickness”.
L172: permafrost top layer --> permafrost table?
Changed.
L176: Kasin or Kashin?
Corrected.
L194-196: here mentioned the seismic measurements, but its result does not find in the following part.
We decided not to include seismic results in the article. The of them are published in
- R. Sadurtdinov, A. G. Skvortsov, A. M. Tsarev, M. S. Sudakova, G. V. Malkova. GEOPHYSICAL METHODS FOR STUDYING ACTIVE LAYER THICKNESS AND ITS PROPERTIES AT CALM SITES (EUROPEAN NORTH) // Environmental and infrastructure integrity in permafrost regions. Proceedings of the Russian conference with international participationon the occasion of the 60th anniversary of the Melnikov permafrost institute. Yakutsk (Russia), september 28–30, 2020 (edited by M.N. Zhelezniak, Dr. sc. (geol. & miner.), V.V. Shepelev, Dr. sc. (geol. & miner.), R.V. Zhang, Dr. sc. (eng.))
L280: B represents Figure6 b?
Corrected.
L336-L339: In the first section (Line336-340), the authors said the velocity was determined through hyperbolas fitting for the not horizontal permafrost table layer, but in the following section (Line341-343) only "multifold" velocity detection method was mentioned, this is confused for readers, and how the authors acquired the changing velocity along a profile
Using multifold data, the EM velocity could be obtained not only by picking maximums on semblance but also by hyperbola fitting.
The MF data semblance could be useful only for symmetric hyperbolas. The hyperbolas of reflected (not diffraction!) wave hodographs obtained using CRG geometry are symmetric only if boundaries have horizontal or close to horizontal shapes.
The hyperbolas of reflected waves are symmetric in CMP data, but in the case of inclined boundaries, the obtained velocity values would be overestimated.
In our case, we obtained velocities from reflected wave hodographs of CRG data using RadexPro software. In this software, the boundary angle, depth and velocity could be adjusted for the theoretical hyperbola to best match the real hyperbola.
L343: miltifold --> multifold
Corrected.
L414, L419, L428: What is the difference among permafrost top layer, permafrost table and permafrost roof?
changed for permafrost table
L453-454: the correlation only found in Kashin site or both site?
At the Kumzha site, there is a correlation between air temperatures and permafrost table depth.
At the Kashin site, there is a correlation between air temperatures and permafrost table depth and between air temperatures and active layer moisture.
Reviewer 3 Report
Dear Authors,
The paper describes a geophysical campaign aiming at characterising the geometry of permafrost regions, exploited through the use of different sensors and approach – features that make the article interesting. On an overall basis, the paper is well written and structured. The background and the research context are both adequately presented as well.
The paper is easy to follow, what I think it is missing is a more detailed description of the experimental campaign in terms of acquisition parameters for the GPR surveys and details on the seismic surveys, in terms of equipment and recording approach.
In addition, I’ve a couple of comments/curiosity:
- it is not clear to me – sorry about this – the geometry of the GPR survey: in the text, line 169, it is written that the distance between acquired samples (representing the inline sampling values, I believe) has been set to 10 cm, while the spacing between parallel profiles (as inferable from Fig. 2) is 10 m. Is this correct?
- Then, as you have acquired a 3D volume (even if aliased due to the excessive crossline spacing), did it provides additional features or evidences?
- From the same figure, it seems that acquisitions were carried out along two orthogonal directions – but no mention of this throughout the text. May I ask why? It might be interesting to see whether a change in the acquisition geometry impacts the results.
Best Regards.
Author Response
Dear Reviewer,
We are grateful for the high rating of our article.
We tried to properly answer all of your questions and correct all of the errors you noted. Point-by-point response are provided below.
We hope you will find all of our corrections suitable.
The paper is easy to follow, what I think it is missing is a more detailed description of the experimental campaign in terms of acquisition parameters for the GPR surveys and details on the seismic surveys, in terms of equipment and recording approach.
The GPR survey technique was not sophisticated and almost everything is described in the “methods” section. The only thing we did not mention was the recording time length. At the Kashin site, it was 100 ns, and at the Kumzha site, it was 300 ns for zero-offset and multifold data.
We decided focus this article on the GPR data, so the seismic survey is outside the scope of this article. Some results can be found in
- R. Sadurtdinov, A. G. Skvortsov, A. M. Tsarev, M. S. Sudakova, G. V. Malkova. GEOPHYSICAL METHODS FOR STUDYING ACTIVE LAYER THICKNESS AND ITS PROPERTIES AT CALM SITES (EUROPEAN NORTH) // Environmental and infrastructure integrity in permafrost regions. Proceedings of the Russian conference with international participationon the occasion of the 60th anniversary of the Melnikov permafrost institute. Yakutsk (Russia), september 28–30, 2020 (edited by M.N. Zhelezniak, Dr. sc. (geol. & miner.), V.V. Shepelev, Dr. sc. (geol. & miner.), R.V. Zhang, Dr. sc. (eng.))
- it is not clear to me – sorry about this – the geometry of the GPR survey: in the text, line 169, it is written that the distance between acquired samples (representing the inline sampling values, I believe) has been set to 10 cm, while the spacing between parallel profiles (as inferable from Fig. 2) is 10 m. Is this correct?
Yes, this is correct. The profiles crossed at 90 degrees only through CALM grid points. The acquisition geometry is 2D geometry, not 3D.
- Then, as you have acquired a 3D volume (even if aliased due to the excessive crossline spacing), did it provides additional features or evidences?
We were focused on CALM goals and unfortunately did not acquire a 3D volume..
- From the same figure, it seems that acquisitions were carried out along two orthogonal directions – but no mention of this throughout the text. May I ask why? It might be interesting to see whether a change in the acquisition geometry impacts the results.
You are totally right about acquisition geometry! We added an additional description to the methods (highlighted in yellow):
The GPR acquisitions were carried out along two orthogonal directions. The profiles crossed CALM grid points, and the distance between parallel profiles was 10 m.
The average mismatch at the Kumzha site was 3 ns, and that at the Kashin site was 0.4 ns, which are appropriate.
Round 2
Reviewer 2 Report
I think the authors has responded to all of my concerns and the manuscript could be accepted at its present form.